# MULTI-CROSSRE
# A Multi-Lingual Multi-Domain Dataset for Relation Extraction

**Elisa Bassignana**⊘   **Filip Ginter**☺   **Sampo Pyysalo**☺
**Rob van der Goot**⊘   **Barbara Plank**⊘▲

⊘Department of Computer Science, IT University of Copenhagen, Denmark
☺TurkuNLP, Department of Computing, University of Turku, Finland
▲MaiNLP, Center for Information and Language Processing, LMU Munich, Germany
`elba@itu.dk`   `figint@utu.fi`

## Abstract

Most research in Relation Extraction (RE) involves the English language, mainly due to the lack of multi-lingual resources. We propose MULTI-CROSSRE, the broadest multi-lingual dataset for RE, including 26 languages in addition to English, and covering six text domains. MULTI-CROSSRE is a machine translated version of CrossRE (Bassignana and Plank, 2022a), with a sub-portion including more than 200 sentences in seven diverse languages checked by native speakers. We run a baseline model over the 26 new datasets and—as sanity check—over the 26 back-translations to English. Results on the back-translated data are consistent with the ones on the original English CrossRE, indicating high quality of the translation and the resulting dataset.

## 1 Introduction

Binary Relation Extraction (RE) is a sub-field of Information Extraction specifically aiming at the extraction of triplets from text describing the semantic connection between two entities. The task gained a lot of attention in recent years, and different directions started to be explored. For example, learning new relation types from just a few instances (few-shot RE; Han et al., 2018; Gao et al., 2019; Sabo et al., 2021; Popovic and Färber, 2022), or evaluating the models over multiple source domains (cross-domain RE; Bassignana and Plank, 2022b,a). However, a major issue of RE is that most research so far involves the English language only.

After the very first multi-lingual work from the previous decade—the ACE dataset (Doddington et al., 2004) including English, Arabic and Chinese—recent work has started again exploring

multi-lingual RE. Seganti et al., 2021 published a multi-lingual dataset, built from entity translations and Wikipedia alignments from the original English version. The latter was collected from automatic alignment between DBpedia and Wikipedia. The result includes 14 languages, but with very diverse relation type distributions: Only English contains instances of all the 36 types, while the most low-resource Ukrainian contains only 7 of them (including the 'no_relation'). This setup makes it hard to directly compare the performance on different languages. Kassner et al., 2021 translated TREx (Elsahar et al., 2018) and GoogleRE,[1] both consisting of triplets in the form (object, relation, subject) with the aim of investigating the knowledge present in pre-trained language models by querying them via fixed templates. In the field of distantly supervised RE, Köksal and Özgür, 2020 and Bhartiya et al., 2022 introduce new datasets including respectively four and three languages in addition to English.

In this paper, we propose MULTI-CROSSRE, to the best of our knowledge the most diverse RE dataset to date, including 27 languages and six diverse text domains for each of them. We automatically translated CrossRE (Bassignana and Plank, 2022a), a fully manually-annotated multi-domain RE corpus, annotated at sentence level. We release the baseline results on the proposed dataset and, as quality check, on the 26 back-translations to English. Additionally, we report an analysis where native speakers in seven diverse languages manually check more than 200 translated sentences and the respective entities, on which the semantic relations are based. MULTI-CROSSRE allows for the investigation of sentence-level RE in the 27 languages included in it, and for direct performance comparison between them. Our contributions are:
① We propose a practical approach to machine-

---

[1] `https://code.google.com/archive/p/relation-extraction-corpus`

In machine learning, support-vector machines (SVMs, also support-vector networks) are supervised learning models with learning algorithms that analyze data used for classification and regression analysis.

Beim maschinellen Lernen sind Support-Vektor-Maschinen (SVMs, auch Support-Vektor-Netzwerke) überwachte Lernmodelle mit Lernalgorithmen, die Daten für Klassifizierungs- und Regressionsanalysen analysieren.

In machine learning, support vector machines (SVMs, also support vector networks) are supervised learning models with learning algorithms that analyse data for classification and regression analysis.

Figure 1: **Example sentence with color-coded entity markup.** From top to bottom: The original English text, its translation to German, and translation back to English. In the first translation step the entity *classification* is not transferred to German. In the second translation step the entity *machine learning* is (wrongly) expanded by a comma—later corrected in our post-processing.

|  | SENTENCES | | | | RELATIONS | | | |
|---|---|---|---|---|---|---|---|---|
|  | train | dev | test | **tot.** | train | dev | test | **tot.** |
| 📰 | 164 | 350 | 400 | 914 | 175 | 300 | 396 | 871 |
| 🏛 | 101 | 350 | 400 | 851 | 502 | 1,616 | 1,831 | 3,949 |
| 🌿 | 103 | 351 | 400 | 854 | 355 | 1,340 | 1,393 | 3,088 |
| 🎵 | 100 | 350 | 399 | 849 | 496 | 1,861 | 2,333 | 4,690 |
| 📖 | 100 | 400 | 416 | 916 | 397 | 1,539 | 1,591 | 3,527 |
| 🤖 | 100 | 350 | 431 | 881 | 350 | 1,006 | 1,127 | 2,483 |
| **tot.** | 668 | 2,151 | 2,446 | **5,265** | 2,275 | 7,662 | 8,671 | **18,608** |

Table 1: **CrossRE Statistics.** Number of sentences and number of relations for each domain.

translate datasets with span-based annotations and apply it to produce MULTI-CROSSRE, the first multi-lingual and multi-domain dataset for RE including 27 languages and six text domains.[2] ② Multi-lingual and multi-domain baselines over the proposed dataset. ③ Comprehensive experiments over the back-translations to English. ④ A manual analysis by native speakers over more than 200 sentences in seven diverse languages.

## 2  MULTI-CROSSRE

**CrossRE** As English base, we use CrossRE (Bassignana and Plank, 2022a),[3] a recently published multi-domain dataset. CrossRE is entirely manually-annotated, and includes 17 relation types spanning over six diverse text domains: artificial intelligence (🤖), literature (📖), music (🎵), news (📰), politics (🏛), natural science (🌿). The dataset was annotated on top of CrossNER (Liu et al., 2021), a Named Entity Recognition (NER) dataset. Table 1 reports the statistics of CrossRE.

**Translation Process** With the recent progress in the quality of machine translation (MT), utilizing machine-translated datasets in training and evaluation of NLP methods has become a standard practice (Conneau et al., 2018; Kassner et al., 2021). As long as the annotation is not span-bound, producing a machine-translated dataset is rather straightforward. The task however becomes more involved for datasets with annotated spans, such as the named entities in our case of the CrossRE dataset, or e.g. the answer spans in a typical question answering (QA) dataset. Numerous methods have been developed for transferring span information between the source and target texts (Chen et al., 2022). These methods are often tedious and in many cases rely on language-specific resources to obtain the necessary mapping. Some methods also require access to the inner state of the MT system, e.g. its attention activations, which is generally not available when commercial MT systems are used.

In this work, we demonstrate a practical and simple approach to the task of machine translating a span-based dataset. We capitalize on the fact that DeepL,[4] a commercial machine translation service very popular among users thanks to its excellent translation output quality, is capable of translating document markup. This feature is crucial for professional translators—the intended users of the service—who need to translate not only the text of the source documents, but also preserve their formatting. In practice, this means that the input of DeepL can be a textual document with formatting (a Word document) and the service produces its translated version with the formatting preserved.

For the CrossRE dataset, we only need to transfer the named entities, which can be trivially encoded as colored text spans in the input documents, where the color differentiates the individual entities. This is further facilitated by the fact that the entities do not overlap in the dataset, allowing for a simple one-to-one id-color mapping. Observing that oftentimes the entities are over-

---

[2] https://github.com/mainlp/CrossRE
[3] Released with a GNU General Public License v3.0.

[4] https://www.deepl.com/translator

| | lang2vec | TRANSLATION (EN → X) | | | | | | | BACK-TRANSLATION (X → EN) | | | | | | | | | | | | | | Δ_BT | Δ_OR |
| | | | | | | | | | EVAL ON BACK-TRANSLATED DATA | | | | | | | EVAL ON ORIGINAL CrossRE DATA | | | | | | | | |
| Language | | 🏠 | 📖 | 🎵 | 📋 | 🏛 | 👟 | avg. | 🏠 | 📖 | 🎵 | 📋 | 🏛 | 👟 | avg. | 🏠 | 📖 | 🎵 | 📋 | 🏛 | 👟 | avg. | | |
|---|---|---|---|---|---|---|---|---|---|---|---|---|---|---|---|---|---|---|---|---|---|---|---|---|
| German | 0.18 | 24.6 | 27.6 | 29.6 | 9.7 | 19.7 | 21.1 | 22.0 | 24.9 | 31.5 | 27.9 | 10.5 | 19.3 | 21.2 | 22.5 | 25.1 | 30.7 | 27.7 | 10.4 | 19.6 | 21.5 | 22.5 | 0.0 | 0.8 |
| Danish | 0.18 | 25.5 | 30.8 | 33.0 | 11.9 | 19.8 | 21.4 | 23.7 | 25.6 | 31.4 | 34.6 | 8.4 | 20.0 | 21.4 | 23.6 | 25.6 | 30.6 | 33.8 | 8.6 | 20.1 | 20.6 | 23.2 | 0.4 | 0.1 |
| Portuguese_BR | 0.18 | 26.2 | 30.7 | 29.2 | 10.7 | 20.0 | 21.2 | 23.0 | 24.9 | 34.7 | 32.1 | 10.1 | 18.2 | 21.5 | 23.6 | 25.3 | 32.5 | 32.5 | 10.1 | 17.9 | 21.4 | 23.3 | 0.3 | 0.0 |
| Portuguese_PT | 0.18 | 28.2 | 32.9 | 31.7 | 10.5 | 20.1 | 22.9 | 24.4 | 24.4 | 34.7 | 28.0 | 10.1 | 19.9 | 21.5 | 23.2 | 25.1 | 34.5 | 28.9 | 10.0 | 19.7 | 22.3 | 23.4 | 0.2 | 0.1 |
| Dutch | 0.19 | 25.8 | 30.9 | 29.3 | 9.7 | 18.5 | 20.7 | 22.5 | 25.0 | 32.1 | 30.3 | 10.5 | 19.9 | 21.6 | 23.2 | 25.7 | 32.2 | 30.3 | 10.7 | 20.4 | 21.8 | 23.5 | 0.3 | 0.2 |
| Ukrainian | 0.21 | 26.7 | 29.1 | 27.5 | 9.0 | 19.4 | 20.4 | 22.0 | 24.8 | 31.4 | 29.9 | 10.4 | 16.1 | 22.5 | 22.5 | 24.6 | 30.9 | 30.5 | 10.8 | 16.2 | 23.3 | 22.7 | 0.2 | 0.6 |
| Swedish | 0.21 | 25.8 | 33.4 | 31.1 | 10.6 | 18.6 | 21.6 | 23.5 | 25.7 | 32.1 | 33.4 | 8.0 | 17.4 | 20.5 | 22.9 | 25.2 | 31.3 | 32.4 | 8.3 | 17.8 | 20.2 | 22.5 | 0.4 | 0.8 |
| Slovenian | 0.22 | 27.0 | 32.3 | 28.1 | 7.9 | 15.0 | 20.1 | 21.7 | 25.3 | 32.4 | 28.4 | 10.5 | 19.8 | 21.1 | 22.9 | 25.1 | 31.3 | 30.2 | 10.1 | 20.0 | 20.2 | 22.8 | 0.1 | 0.5 |
| Italian | 0.22 | 27.1 | 32.5 | 31.3 | 12.8 | 19.1 | 22.3 | 24.2 | 26.3 | 34.6 | 32.0 | 11.3 | 19.9 | 19.7 | 24.0 | 26.7 | 34.3 | 31.5 | 11.3 | 20.2 | 20.0 | 24.0 | 0.0 | 0.7 |
| Romanian | 0.23 | 26.5 | 33.0 | 30.2 | 10.3 | 16.6 | 21.3 | 23.0 | 24.0 | 33.7 | 29.8 | 10.8 | 20.7 | 19.4 | 23.1 | 24.3 | 30.5 | 30.4 | 10.8 | 20.0 | 19.2 | 22.5 | 0.6 | 0.8 |
| Bulgarian | 0.23 | 28.1 | 34.4 | 27.2 | 9.0 | 20.4 | 20.9 | 23.3 | 24.3 | 31.5 | 29.2 | 10.8 | 19.1 | 21.4 | 22.7 | 24.3 | 31.1 | 30.9 | 10.9 | 19.0 | 21.5 | 22.9 | 0.2 | 0.4 |
| French | 0.23 | 29.6 | 33.5 | 32.3 | 11.3 | 19.3 | 23.5 | 24.9 | 25.5 | 33.5 | 31.4 | 11.2 | 19.8 | 21.8 | 23.9 | 25.5 | 32.1 | 31.2 | 10.9 | 20.1 | 21.7 | 23.6 | 0.3 | 0.3 |
| Slovak | 0.23 | 23.1 | 32.7 | 28.2 | 9.2 | 18.6 | 18.2 | 21.7 | 24.4 | 32.6 | 31.6 | 10.2 | 19.2 | 19.8 | 23.0 | 24.1 | 33.6 | 31.7 | 10.3 | 17.8 | 20.1 | 22.9 | 0.1 | 0.4 |
| Indonesian | 0.24 | 26.0 | 34.6 | 33.2 | 9.6 | 19.7 | 20.7 | 24.0 | 25.2 | 32.9 | 32.6 | 9.7 | 16.9 | 20.9 | 23.0 | 26.1 | 32.9 | 32.4 | 9.8 | 16.5 | 20.7 | 23.1 | 0.1 | 0.2 |
| Latvian | 0.25 | 24.8 | 32.3 | 25.0 | 11.0 | 15.9 | 19.1 | 22.0 | 24.3 | 32.6 | 27.6 | 8.7 | 18.8 | 20.5 | 22.1 | 24.4 | 30.9 | 28.7 | 8.5 | 19.1 | 20.5 | 22.0 | 0.1 | 1.3 |
| Spanish | 0.27 | 27.6 | 32.2 | 29.9 | 9.7 | 19.2 | 22.5 | 23.5 | 24.5 | 32.4 | 29.1 | 9.2 | 19.5 | 23.9 | 23.1 | 24.6 | 31.9 | 28.6 | 9.5 | 20.2 | 23.3 | 23.0 | 0.1 | 0.3 |
| Hungarian | 0.27 | 22.4 | 28.9 | 26.0 | 8.5 | 19.2 | 18.4 | 20.6 | 21.2 | 31.0 | 28.5 | 8.6 | 18.5 | 21.2 | 21.5 | 22.2 | 30.2 | 29.1 | 8.5 | 19.3 | 21.3 | 21.8 | 0.3 | 1.5 |
| Greek | 0.27 | 28.3 | 33.3 | 31.8 | 9.1 | 20.3 | 22.7 | 24.2 | 24.1 | 30.7 | 32.9 | 11.2 | 18.6 | 19.8 | 22.9 | 24.7 | 31.9 | 33.6 | 10.9 | 19.2 | 20.8 | 23.5 | 0.6 | 0.2 |
| Estonian | 0.27 | 23.4 | 29.3 | 27.4 | 8.3 | 17.1 | 19.0 | 20.8 | 22.7 | 31.8 | 29.2 | 8.5 | 15.8 | 19.4 | 21.2 | 23.8 | 30.6 | 30.4 | 8.5 | 16.4 | 18.4 | 21.3 | 0.1 | 2.0 |
| Lithuanian | 0.27 | 26.2 | 31.5 | 26.3 | 9.9 | 18.9 | 16.2 | 21.5 | 24.5 | 31.3 | 26.4 | 10.8 | 18.8 | 21.4 | 22.2 | 25.3 | 30.0 | 27.6 | 10.3 | 18.6 | 21.2 | 22.2 | 0.0 | 1.1 |
| Polish | 0.27 | 24.6 | 34.3 | 28.7 | 10.4 | 19.5 | 19.9 | 22.9 | 24.4 | 31.6 | 27.9 | 9.7 | 16.6 | 20.4 | 21.8 | 24.5 | 30.9 | 28.6 | 9.6 | 16.6 | 20.8 | 21.8 | 0.0 | 1.5 |
| Finnish | 0.28 | 22.9 | 30.2 | 24.7 | 8.8 | 17.0 | 18.1 | 20.3 | 21.4 | 29.5 | 27.1 | 8.8 | 17.4 | 20.5 | 20.8 | 24.9 | 34.7 | 32.1 | 10.1 | 18.2 | 21.5 | 23.6 | 2.8 | 0.3 |
| Czech | 0.29 | 25.0 | 30.1 | 28.4 | 10.1 | 19.4 | 18.1 | 21.8 | 23.8 | 30.8 | 29.0 | 9.8 | 20.2 | 19.6 | 22.2 | 24.4 | 31.9 | 29.5 | 9.7 | 19.6 | 20.0 | 22.5 | 0.3 | 0.8 |
| Chinese | 0.30 | 22.2 | 33.4 | 25.0 | 9.0 | 20.1 | 18.7 | 21.4 | 23.1 | 28.4 | 27.1 | 9.5 | 18.9 | 22.0 | 21.5 | 23.8 | 28.7 | 27.4 | 9.9 | 18.7 | 21.3 | 21.6 | 0.1 | 1.7 |
| Turkish | 0.38 | 23.8 | 29.4 | 26.7 | 10.6 | 20.4 | 18.2 | 21.5 | 23.4 | 23.2 | 28.4 | 9.3 | 17.6 | 19.1 | 20.2 | 24.5 | 23.2 | 29.8 | 9.2 | 17.9 | 20.3 | 20.8 | 0.6 | 2.5 |
| Japanese | 0.41 | 22.6 | 29.2 | 20.1 | 8.9 | 19.5 | 12.9 | 18.9 | 21.1 | 27.4 | 21.7 | 8.0 | 16.1 | 15.2 | 18.3 | 20.5 | 27.9 | 23.4 | 8.1 | 16.1 | 16.2 | 18.7 | 0.4 | 4.6 |

Table 2: **MULTI-CROSSRE Baseline Results.** Macro-F1 scores of the baseline model ordered by increasing lang2vec distance from English. $\Delta_{BT}$: delta between back-translated and original evaluation when model trained on back-translated data. $\Delta_{OR}$: delta between model trained on back-translated data and on original CrossRE data when evaluated on original CrossRE English.

| | 🏠 | 📖 | 🎵 | 📋 | 🏛 | 👟 | avg. |
|---|---|---|---|---|---|---|---|
| English | 20.8 | 36.4 | 30.7 | 10.1 | 20.0 | 21.6 | 23.3 |

Table 3: **CrossRE Baseline Results.** Macro-F1 scores of the RC baseline over the original CrossRE English dataset.

extended by a punctuation symbol during translation, the only post-processing we apply is to strip from each translated entity any trailing punctuation not encountered in the suffix of the original named entity. The process is illustrated in Figure 1, with details about two typical issues with this approach (later analysed in Section 4).[5]

## 3 Experiments

**Model Setup** In order to be able to directly compare our results with the original CrossRE baselines on English, we follow the model and task setup used by Bassignana and Plank, 2022a. We perform Relation Classification (Han et al., 2018; Baldini Soares et al., 2019; Gao et al., 2019), which consists of assigning the correct relation types to the ordered entity pairs which are given as semantically connected. The model follows the current state-of-the-art architecture by Baldini Soares et al., 2019 which augments the

sentence with four entity markers $e_1^{start}$, $e_1^{end}$, $e_2^{start}$, $e_2^{end}$ surrounding the two entities. Following Zhong and Chen (2021) the entity markers are enriched with information about the entity types. The augmented sentence is then passed through a pre-trained encoder (XLM-R large; Conneau et al., 2020), and the classification made by a linear layer over the concatenation of the start markers $[\hat{s}_{e_1^{start}}, \hat{s}_{e_2^{start}}]$. We run all our experiments over five random seeds. See Appendix A for reproducibility and hyperparameters settings.

**Results** The original CrossRE study reports the baseline experiments by using the mono-lingual BERT (Devlin et al., 2019) language encoder. In order to be able to compare the original baseline with the results on our MULTI-CROSSRE dataset, we re-run the English experiments by using the multi-lingual XLM-R large (Conneau et al., 2020) language encoder, and report the results in Table 3.

In Table 2 we report the results of our experiments over MULTI-CROSSRE. The left-most columns are the results of the models trained and evaluated over the translated data (from English to language X). As a sanity check, we back-translated the data from each of the 26 new languages to English (from language X to English). We train and evaluate new models on this data in the middle columns. Finally, on the right-most

---

[5]The overall translation process cost is ≈ 60€.

columns we evaluate the same models—trained on back-translated data—over the original CrossRE test sets. We sort the languages by increasing distance to English, computed as the cosine distance between the syntax, phonology and inventory vectors of lang2vec (Littell et al., 2017).

For our analysis we consider the average of the six domains.[6] Our scores on the translated data reveal a relatively small drop in respect to the English baseline in Table 3. The difference range goes from an improvement of $+1.6$ Macro-F1 points on French, to a maximum drop of $-4.4$ on Japanese—which has the largest lang2vec distance with respect to English (0.41). The results of the models trained on the back-translated data present essentially the same trend between evaluating on the back-translations and on the original CrossRE English data—with a Pearson's correlation coefficient of 0.88—confirming the high quality of the proposed translation. The only exception if Finnish, with a difference of 2.8 points between the two evaluations. All the other languages report a smaller difference in a range between 0.0 and 0.6. The lang2vec distance is not informative of the quality of the individual translations (Pearson's correlation $-0.59$). However, other factors should be taken into account, e.g. the language model performances on each individual language.

## 4 Manual Analysis

We performed a manual analysis for further inspecting the quality and usability of MULTI-CROSSRE for studying multi-lingual RE. We manually checked 210 sentences from a diverse set of seven languages, including one North Germanic (Danish), one Uralic (Finnish), one West Slavic (Czech), two Germanic (German and Dutch), one Latin (Italian), and one Japonic (Japanese). For each of them, native speakers annotated the following: ① In how many sentences is the overall meaning preserved? ② How many entities are transferred to language X? ③ How many entities are correctly translated? ④ How many entities are marked with the correct entity boundaries?

We annotated 30 sentences for each language. Table 4 reports the statistics of our analysis. Overall, we find a surprisingly high quality of entity translations (96% are judged as correct by our

---

[6]Bassignana and Plank, 2022a discuss the lower scores of news (▤) attributing them to the data coming from a different data source and the fewer amount of relation instances with respect to the other domains.

| Language | Sent. Transl. | # entities | Ent. Transl. | Ent. Bound. |
|---|---|---|---|---|
| English | 30 | 160 | - | - |
| Czech | 28 | 158 | 152 | 143 |
| Danish | 27 | 158 | 143 | 136 |
| Dutch | 28 | 158 | 156 | 141 |
| Finnish | 30 | 150 | 141 | 137 |
| German | 27 | 151 | 148 | 139 |
| Italian | 29 | 160 | 157 | 152 |
| Japanese | 19 | 150 | 145 | 82 |

Table 4: **Statistics of the Manual Analysis.** At the top, total amount of original English sentences and annotated entities within them. Below, for each sample set, amount of correct instances in the four categories of sentence translation, number of entities, entity translations, and entity boundaries.

human annotators). Out of the seven languages, Japanese is the one suffering the most by the translation process and, as we discussed above, this is reflected in the lowest scores in Table 2. Some entities are not transferred. These are mostly due to compounds typical for some languages. For example, the English snippet "the *Nobel* laureate" (where only *Nobel* is marked as entity), is translated to Danish as "nobelpristageren", and to Dutch as "Nobelprijswinnaar". In Italian, which in this regard behaves more similarly to English, all the entities are correctly transferred. In Appendix B we report the total per-language percentages of transferred entities and relations. Regarding the entity translations and the entity boundaries, the latter is a bigger challenge for the translation tool, often including surrounding function words—e.g. the writer *Pat Barker* in Danish is extended to the entity *Pat Barker er*. These could easily be post-processed, but since the Relation Classification model relies on the injected entity markers, it is not much influenced by this type of error (see baseline discussion in Section 3).

## 5 Conclusion

We introduce MULTI-CROSSRE, the most diverse RE dataset to date, including 26 languages in addition to the original English, and six text domains. The proposed span-based MT approach could be easily applied to similar cases. We report baseline results on the proposed resource and, as quality check, we back-translate MULTI-CROSSRE to English and run the baseline model again over it. Our manual analysis reveals that the higher challenge during the translation is transferring the correct entity boundaries. However, given the model architecture, this does not influence the scores.

## Acknowledgments

We thank the MaiNLP/NLPnorth group for feedback on an earlier version of this paper, and ITU's High-performance Computing cluster for computing resources.

EB and BP are supported by the Independent Research Fund Denmark (Danmarks Frie Forskningsfond; DFF) Sapere Aude grant 9063-00077B. BP is supported by the ERC Consolidator Grant DIALECT 101043235. FG and SP were supported by the Academy of Finland.

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

# Appendix

## A Reproducibility

We report in Table 5 the hyperparameter setting of our RC model (see Section 3). All experiments were ran on an NVIDIA® A100 SXM4 40 GB GPU and an AMD EPYC™ 7662 64-Core CPU.

| Parameter | Value |
|---|---|
| Encoder | `xlm-roberta-large` |
| Classifier | 1-layer FFNN |
| Loss | Cross Entropy |
| Optimizer | Adam optimizer |
| Learning rate | $2e^{-5}$ |
| Batch size | 32 |
| Seeds | 4012, 5096, 8878, 8857, 9908 |

Table 5: **Hyperparameters Setting.** Model details for reproducibility of the baseline.

| Language | % Entities | % Relations |
|---|---|---|
| German | 96.7 | 91.4 |
| Danish | 97.5 | 93.9 |
| Portuguese_BR | 99.8 | 99.5 |
| Portuguese_PT | 99.8 | 99.6 |
| Dutch | 98.5 | 95.8 |
| Ukrainian | 99.1 | 97.7 |
| Swedish | 97.6 | 94.1 |
| Slovenian | 99.1 | 98.0 |
| Italian | 99.8 | 99.5 |
| Romanian | 98.8 | 96.7 |
| Bulgarian | 99.5 | 98.9 |
| French | 99.6 | 99.4 |
| Slovak | 99.2 | 98.1 |
| Indonesian | 99.8 | 99.5 |
| Latvian | 99.4 | 98.6 |
| Spanish | 99.3 | 98.3 |
| Hungarian | 98.2 | 95.8 |
| Greek | 98.8 | 98.0 |
| Estonian | 97.9 | 94.6 |
| Lithuanian | 99.4 | 98.8 |
| Polish | 99.4 | 98.6 |
| Finnish | 96.0 | 90.7 |
| Czech | 99.0 | 98.0 |
| Chinese | 99.3 | 98.4 |
| Turkish | 99.4 | 98.5 |
| Japanese | 94.9 | 88.9 |

Table 6: **Transferred Entities and Relations.** Percentages of entities and of relations transferred during the translation process for each language.

## B Per-language Analysis

In table 6 we report the percentages of entities which are transferred during the translation process from the original English to language X, and the percentage of relations which do not involve missing entities (i.e. are transferred during the translation process).