# OpenReview forum: "Multi-CrossRE A Multi-Lingual Multi-Domain Dataset for Relation Extraction"
_NoDaLiDa/2023/Conference — NoDaLiDa 2023_

### Official Review · Reviewer_1JN1 · 2023-03-08
**A good dataset for multilingual Relation Extraction**

**Rating:** 7
**Confidence:** 5

**Review:**

The paper presents the results of RE for a known set containing the relation annotation in English and automatically translated to seven languages. DeepL, a commercial machine translation service, performs the translation and back-translation. The sanity checks, the automatic evaluation, and the manual evaluation are all sound. The paper has an appendix where all experiments and the hyperparameters are well-explained, enabling the reproducibility of the work. The translated set and the experiments are an excellent addition to the multilingual RE research.
I have two comments that are easily addressable :
1. The definition of relation extraction needs to be more accurate. The authors state: "Relation Extraction (RE) is a sub-field of Information Extraction specifically aiming at the extraction of triplets from text describing the semantic connection between two entities." What they present is the task of binary relation extraction. The extracted relations can hold between more than two entities. Also, in the given dataset, the entities should be inside the sentence boundaries. But even the binary task can be made more general, as the two entities can be in the same article, not necessarily in the same sentence. Please, add these restrictions to your definition (entities in the same sentence and binary RE task).
2. The dataset assumes no overlap in the sentence between the entity span. However, this is only sometimes the case; in many industrial applications, the entity span for which we want to perform RE overlaps. Can the translation framework you develop be adapted to a dataset containing such cases?

**Paper Type:**

Short paper

---

### Official Review · Reviewer_peKA · 2023-03-14
**Neat approach to creating RE datasets in multiple languages**

**Rating:** 9
**Confidence:** 5

**Review:**

The paper describes a multilingual dataset (27 languages) for relation extraction. The dataset has been created by automatically translating an English dataset into all the other languages and transferring the annotation. (Here, in order to transfer the annotattion spans, the approach relies on a translation engine capable of aligning markup between the original text and the translation.) There are some checks on the reliability of this process: sentences are translated back into English and the results compared to the original dataset, and for 7 of the languages manual inspection is done on a subset. Baseline results for RE in the 27 languages are computed. There are some analysis on the effect of the linguistic distance on the quality of the results; as can be expected, the worst results are for the typologically most distant language (Japanese).

All in all, this is extremely neat and should definitely be accepted as a short. The methods are simple and effective, and there is enough analysis of the strengths and weaknesses for an acceptance. The approach seems applicable to a wide range of simple markup-based IE tasks.

Typo: “concept aggregations” -> compounds. Please note that the difference between English and e.g. Dutch or Danish in whether whitespace is used in compounds or not is simply a matter of orthographical conventions, not a typological difference.


**Paper Type:**

Short paper

---

### Official Review · Reviewer_JNod · 2023-03-14
**Relation Extraction, dataset, multi-lingual, multi-domain**

**Rating:** 7
**Confidence:** 2

**Review:**

The paper introduces a new multilingual relation extraction dataset in terms of its development and evaluation.  The paper is well written and fairly clear even to someone coming from outside the field.  The presented evaluation using native speakers of the languages is especially valuable.  The English used is adequate, but the paper has some minor typographical issues (orphaned lines).  All in all the manuscript depicts a competent work presented clearly.

**Paper Type:**

Long paper

---

### Decision · Program_Chairs · 2023-03-17

Accept